# ELICITATION INFERENCE OPTIMIZATION FOR MULTI-PRINCIPAL-AGENT ALIGNMENT

## ABSTRACT

In multi-principal-agent alignment scenarios including governance, markets, conflict resolution, and AI decision-making, it is infeasible to elicit every principal's view on all perspectives relevant to an agent's decisions. Elicitation inference optimization (EIO) aims to minimize the $n$ elicitations needed to approximate $N$ principal's views across $K$ perspectives. In this work, we demonstrate an EIO approach where data efficiency ($NK/n$) increases with scale. We introduce STUMP: an elicitation inference model which integrates a large language model with a latent factor model to enable learning transfer across samples, contexts, and languages. We characterize STUMP's performance on a set of elicitation primitives from which scalable elicitation (sampling) protocols can be constructed. Building from these results, we design and demonstrate two elicitation protocols for STUMP where, surprisingly, data efficiency scales like $O(n)$ in the number of elicitations $n$. In other words, the number of elicitations needed per principal remains constant even as the number of perspectives and principals grows. This makes it possible to approximate complex, high-dimensional preference signals spanning principal populations at scale.

## 1 INTRODUCTION

**The principal-agent problem** involves aligning agent decisions with principal interests. A challenge is creating situations where agent choices are sufficiently influenced by signals containing principal preferences. With a single principal, high-complexity preference signals can be elicited directly via open-ended interaction. Multi-principal-agent scenarios can involve large populations of principals and powerful agents such as: governments & citizens Giger and Lefkofridi (2014); Gabriel (2020), firms & customers Roberts and Grover (2012), peacekeepers & conflict parties United Nations (2012), existing AI systems & impacted populations Prabhakaran et al. (2022), and potentially even transformative AI & humanity Russell et al. (2015); Christiano et al. (2017). As the number of principals grows, and the domain of agent decisions becomes open-ended, directly eliciting the preference of all principals on all relevant perspectives becomes unfeasible [A.1]. As a result, lower-complexity forms of elicitation like ballot voting (ie. for governments) and price signals (ie. for firms) are used to learn preferences. While clearly effective—a basis of democracy and the economy—these approaches drastically simplify real preferences. For example, they do not allow citizens (principals) to express *what* they would like a government (agent) to do, or *why*, only *if* they support predefined options.

**Elicitation inference optimization (EIO)** aims to decrease the amount of direct elicitation needed to recover a preference signal (enabling the use of more complex, higher-dimension preferences; e.g. in natural language). Consider an $N \times K$ matrix $\Theta$ where rows correspond to $N$ principals, columns correspond to $K$ perspectives, and every element captures a principal-perspective relationship. *The goal of elicitation inference optimization* is to obtain a sufficient approximation of $\Theta$ with a minimal elicitation budget by directly sampling some elements and inferring the rest. Thus, EIO involves combining a) a sparse elicitation (sampling) protocol with, b) an elicitation inference model.

**Closed-ended surveys simplify EIO by constraining the set of relevant perspectives** to a predefined set — typically, with $K << N$. Matrix sampling techniques elicit responses from each participant on a subset of perspectives selected randomly Shoemaker (1973), heuristically Raghunathan and Grizzle (1995), or dynamically such that inference accuracy is adaptively optimized

Gonzalez and Eltinge (2008); Zhang et al. (2020). Inference models exploit the low-rank-ness of $\Theta$ via matrix factorization and other collaborative filtering techniques Zhang et al. (2020); Sengupta et al. (2021); Oliveira et al. (2021).

**Collective response systems (CRS) enable high-complexity signals** by allowing the set of relevant perspectives to be open-ended: neither limited in number nor requiring pre-definition Ovadya (2022). In a CRS, participants contribute open-ended perspectives in the context of a question, prompt, or conversation and respond to subsets of perspectives contributed by others – typically, yielding $K \sim O(N)$. Inspired by preference models Thurstone (1927) and conjoint analysis Green and Srinivasan (1978), one class of approaches elicits randomized pair-choice votes Konya and Slodov (2015) and uses hierarchical probit-like models for inference Salganik and Levy (2015). Other approaches elicit agreement votes to learn absolute human-perspective relationships. Polis samples agreement votes semi-randomly, prioritizes votes that improve clustering, and uses mean imputation to support dimensionality reduction Small et al. (2021). Additional work bridges these approaches by randomly sampling both pair-choice and agreement votes; inferring missing votes by modeling both vote types with a single utility matrix learned via regularized matrix completion Bilich et al. (2019).

**Previous approaches to EIO for CRS have been limited in scope** to single samples where the set of relevant perspectives all correspond to the same context (ie. the same question or prompt) Salganik and Levy (2015); Konya and Slodov (2015); Small et al. (2021); Bilich et al. (2019). This means votes must be elicited from every person for every new context. As a result, to approximate some $\Theta$ spanning an arbitrary number of contexts, the number of direct elicitations needed grows proportional to the number of elements in $\Theta$ [A.2] – data efficiency does not increase with scale. However, previous approaches do not fully leverage all available data: learning from one context is not transferred to support inference in other contexts, and the information contained in the perspective text is not used at all. *Can an EIO approach that better leverages available data, enable increasing data efficiency with scale?*

**In this work** we introduce an approach to EIO for CRS which becomes increasingly data efficient as the amount of data elicited grows. First, we introduce a novel elicitation inference model – STUMP – which better leverages available data by integrating a pre-trained LLM with a latent factor model. Second, we characterize STUMP's performance across a range of elicitation primitives from which arbitrarily scalable elicitation (vote sampling) protocols can be constructed. Finally, building on these results, we design and demonstrate two scalable elicitation protocols where STUMP can infer with meaningful accuracy while data efficiency increases linearly with scale.

## 2 PROBLEM SETUP

**Elicitation form**. In a collective response system (CRS) participants respond to prompts with open-ended perspectives and vote on perspectives submitted by others. Let $H$ be the set of $N$ human participants, $P$ be the set of $K$ perspectives, and $\Theta$ be the corresponding $N \times K$ human-perspective matrix. Both agreement and pair-choice votes may be elicited on perspectives. In agreement exercise $e^a_{ij}$, participant $i$ is asked if they agree with perspective $j$. In pair choice exercise $e^c_{ijk}$, participant $i$ is asked if they prefer perspective $j$ or $k$. We denote a set of exercise data $E = \{e^a_{ij}, e^c_{ijk} \mid i \in H; j, k \in P\}$.

**Elicitation inference** involves training a model on available data and predicting all missing elements in a target $\Theta$. We choose binary agreement as the human-perspective relationship for our target $\Theta$. Inference performance is probed experimentally by training on a subset of exercises including both vote types $E_t$, and computing prediction accuracy on a validation set of only agreement votes $E_v$.

**Data efficiency** in EIO can be quantified as the ratio of a) the number of votes contained in the target $\Theta$ and b) the number of votes directly elicited via a sampling protocol which a model needs to be trained on to meaningfully approximate $\Theta$. We denote the *data efficiency* to achieve inference accuracy $acc$ as:

$$\beta_{acc} = \frac{n(\Theta)}{n(E_t)} = \frac{n(H) \times n(P)}{n(E_t)} = \frac{NK}{n_t}$$

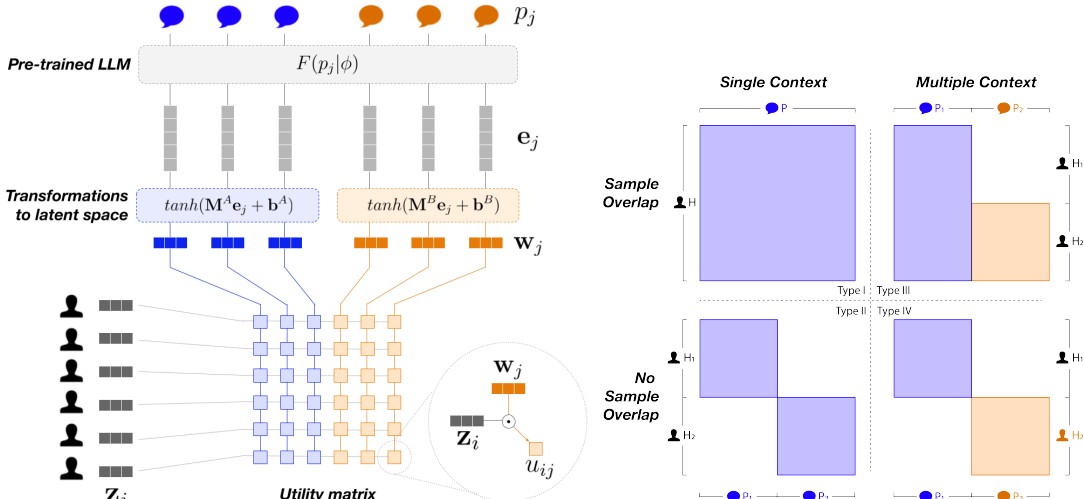

Figure 1: Diagram of STUMP. Universal model elements shown in grey. Perspectives and context-specific model elements colored by context.

Figure 2: Elicitation primitives. Color boxes indicate sparse vote data available for training. Colors denote context.

where $n(X)$ denotes the size of set $X$ and $n_t \equiv n(E_t)$. We refer to the rate data efficiency changes with more elicited votes ($\partial\beta_{acc}/\partial n_t$) as *data leverage*. EIO becomes more data efficient with scale if data leverage is positive for arbitrarily large $n_t$ above a sufficient threshold of accuracy. To achieve this, the challenge is to combine: a) a scalable elicitation (sampling) protocol where $NK$ grows faster than $n_t$, and b) an inference model that sustains meaningful inference accuracy at scale using the data the protocol generates.

## 3 ELICITATION INFERENCE MODEL

**Elicitation inference (EI) is analogous to collaborative filtering (CF)** in recommenders, except instead of items people may prefer, there are *perspectives* people may agree with. A standard approach to CF is to learn a low-rank approximation of a user-item preference matrix by optimizing an objective function against observed preferences (Johnson, 2014). Bilich et al. (2019) demonstrated a similar approach for EI on CRS data. They use regularized matrix completion with a low-rank-promoting nuclear norm prior similar to Fazel et al. (2001) to learn a utility matrix that predicts votes. We refer to this as the *nuclear norm* model (NN), and adopt it as a baseline (see A.3 for technical details).

**Latent factor models** (LFMs) learn vectors for each user $\mathbf{z}_i$ and item $\mathbf{w}_j$ which embed them in the same latent space, and describe the preference of user $i$ for item $j$ as a function of $u_{ij} = \mathbf{z}_i \cdot \mathbf{w}_j$ (Johnson, 2014). Known item features (i.e., director of a movie) can be used to improve performance by projecting features into latent space as a component of item embeddings (Zhang et al., 2021). Relevant features can also be extracted from item content (i.e., a song's time sequence) by a link to latent space embeddings (van den Oord et al., 2013).

**We consider EI on text-based perspectives.** Collaborative Topic Ranking (CTR) uses information in text by learning a topic model on item texts and equating topic feature vectors to latent space embeddings (Xin et al., 2015). Pre-trainable large language models (LLMs) (Cer et al., 2018; Devlin et al., 2018; Yang et al., 2019) learn to encode information from text into embeddings suitable for a wide range of downstream tasks (Men; Hao et al., 2020; Schick and Schütze, 2021). Our approach to using the information in perspective texts is to start with a pre-trained LLM, then similar to Zhang et al. (2021), learn a transformation of text embeddings into latent space. A key challenge in CF is maintaining alignment between latent spaces learned from non-overlapping data (Cremonesi and Quadrana, 2014). Sharing the same embedding model and transformation across perspectives enforces alignment even in the absence of overlapping data (see section 4.3).

**Cross-Domain recommendation** (CDR) involves aligning latent spaces across items in different domains like movies and songs (Zang et al., 2022). One class of approaches uses information like tags that can be shared across domains (Sahu et al., 2018). Others focus on aligning latent space distributional properties (Liu et al., 2021).

**In EI the analog to CDR is multi-context inference.** How one votes on a perspective depends on the context it was elicited in. Take the perspective "*I feel happy.*" Consider two scenarios, where it was elicited by asking "*What would you say if you saw your parents smile?*" or "*What wouldn't you say if you saw your parents smile?*". One would expect nearly opposite voting behavior. But, if the same embedding model and transformation are shared in both contexts, then predictions will be the same. Enabling multi-context inference requires relaxing this condition. In the spirit of Sahu et al. (2018) our approach is to share the same LLM across contexts, but learn a unique transformation into latent space for each context.

**Our model.** We formally introduce our model, the Semantic Transfer Utility Model of Participants: STUMP. We denote the event that participant $i$ reports agreement with perspective $j$ by $a_{ij}$, and disagreement by $d_{ij}$. We denote the event that participant $i$ reports preference for perspective $j$ over perspective $k$ by $c_{ijk}$. We assume a logit utility model of choice similar to Johnson (2014); Bilich et al. (2019) where participant $i$ has a latent utility $u_{ij}$ corresponding to perspective $j$ and voting probabilities are given by $p(a_{ij}|u_{ij}) = \sigma(u_{ij})$, $p(d_{ij}|u_{ij}) = \sigma(1 - u_{ij})$ and $p(c_{ijk}|u_{ij}) = \sigma(u_{ij} - u_{ik})$, where $\sigma$ denotes the logistic function. The utility $u_{ij}$ for participant $i$ and perspective $j$ is constructed as an LFM:

$$u_{ij} = \mathbf{z}_i \cdot \mathbf{w}_j$$

where $\mathbf{z}_i$ is a latent embedding for participant $i$, and $\mathbf{w}_j$ is a latent embedding for perspective $j$. To use information in perspective text, we employ a pre-trained LLM to generate text embeddings. We denote the embedding of the $j^{th}$ perspective text $p_j$ by model $F$ with parameters $\phi$, as:

$$\mathbf{e}_j = F(p_j|\phi).$$

Text embeddings are mapped into latent space by context-specific transformations:

$$\mathbf{w}_j = tanh(\mathbf{M}^q \mathbf{e}_j + \mathbf{b}^q)$$

where $\mathbf{M}^q$ and $\mathbf{b}^q$ are the transformation weight and bias for context $q$ (see Figure 1). Finally, let $A = \{i, j | a_{ij}\}$, $D = \{i, j | d_{ij}\}$, and $C = \{i, j, k | c_{ijk}\}$. The log loss is defined by:

$$L = -(\sum_{i,j,k \in C} log(\sigma(u_{ij} - u_{ik})) + \sum_{i,j \in A} log(\sigma(u_{ij})) + \sum_{i,j \in D} log(1 - \sigma(u_{ij}))) + \lambda_m \|\mathbf{M}\|_F$$

where $\|.\|_F$ denotes the Frobenius norm of a matrix, and $\lambda_m$ is a hyper-parameter controlling regularization strength. The participant embedding vectors $\mathbf{z}$ and the set of transformation parameters for each context $\{\mathbf{M}^q, \mathbf{b}^q\}$ are learned by minimizing the log-loss for observed data. Optionally, the LLM parameters $\phi$ may be fine-tuned through the same minimization.

## 4 ELICITATION PRIMITIVES

Our strategy for elicitation protocol design begins with characterizing STUMP's performance across a range of elicitation primitives from which arbitrarily scalable protocols can be constructed. To this end, we explore four types of elicitation primitives arising from two orthogonal data set properties: *context homogeneity* and *sample overlap*.

**Context homogeneity.** Perspectives and votes are elicited under some context — for example, in response to a question. Thus, a data set (or $\Theta$) may correspond to one or more contexts. We denote the set of perspectives corresponding to the $q^{th}$ context as $P^q \subseteq P$.

**Sample overlap.** Vote data $E$ may come from one or more samples, where each $s^{th}$ sample involves a subset of humans $H_s \subseteq H$ and a subset of perspectives $P_s \subseteq P$. Two samples are said to have *no overlap* when they share neither humans nor perspectives, ie $H_1 \cap H_2 = \emptyset$ and $P_1 \cap P_2 = \emptyset$. A sample with *overlap* may correspond to a data set with only a single sample, or with multiple samples where either humans or perspectives are shared, ie $H_1 \cap H_2 \neq \emptyset$ or $P_1 \cap P_2 \neq \emptyset$.

Permuting these two properties gives four types of elicitation primitives (Figure 2, Table 1):

Table 1: Properties of elicitation primitives.

| Type | Context | Sample | Definition |
|------|---------|--------|------------|
| **I** | Single | Overlap | $P^q = P$ and $H_1 \cap H_2 \neq \emptyset$ or $P_1 \cap P_2 \neq \emptyset$ |
| **II** | Single | No overlap | $P^q = P$ and $H_1 \cap H_2 = P_1 \cap P_2 = \emptyset$ |
| **III** | Multiple | Overlap | $P^q \neq P$ and $H_1 \cap H_2 \neq \emptyset$ or $P_1 \cap P_2 \neq \emptyset$ |
| **IV** | Multiple | No overlap | $P^q \neq P$ and $H_1 \cap H_2 = P_1 \cap P_2 = \emptyset$ |

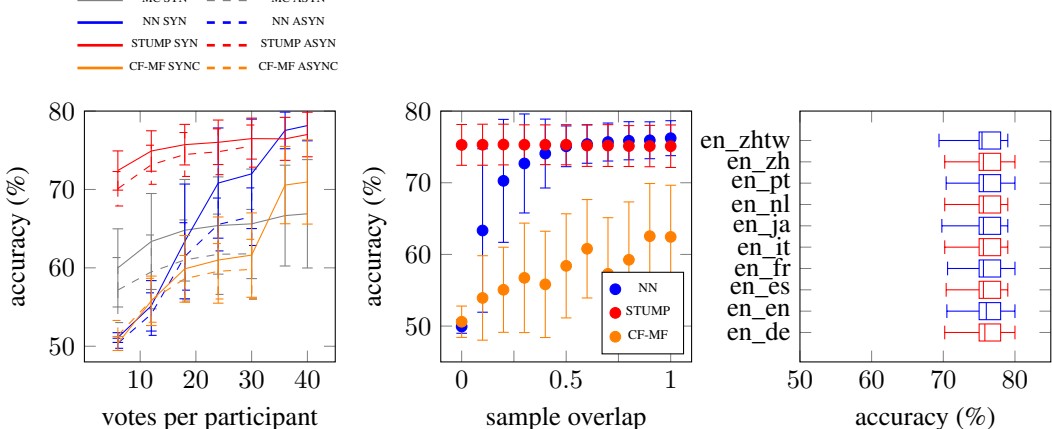

Figure 3: Type I inference accuracy vs votes per participant.

Figure 4: Type I/II inference accuracy vs sample overlap.

Figure 5: Type II inference accuracy by language pair.

## 4.1 ELICITATION PRIMITIVE EXPERIMENTS

**Baselines.** We use four prediction baselines to contextualize STUMP's performance: a) a *Random* baseline where votes are predicted randomly with uniform probability, b) a *Most Common* (MC) baseline where votes are predicted to be the most common choice observed for each perspective, c) the *nuclear norm* (NN) model from Bilich et al (Bilich et al., 2019) [A.3] which represents a standard approach to CF and is the only prior work we are aware of focused on single-vote-level inference for CRS, and d) the classic *matrix factorization* (CF-MF) model widely used in recommendation systems (Koren et al., 2009).

**Data and implementation**. Data used for experiments was collected as part of regular load testing for an existing CRS product and de-identified prior to use in this study. Data came from 534 participants solicited through online research panels. We use data corresponding to seven questions [A.4] asked during the CRS process, chosen for their nuance and presumed hard-to-predict nature. Each question is treated as a unique context, meaning only perspectives and votes elicited in response to the same question share the same context transformation weights. The full dataset used consists of 172,242 votes across 3,703 perspectives made by 534 humans. Experiments are constructed by partitioning exercise data $E$ into training and validation sets ($E_t$ and $E_v$). In each of the primitives described below, the inference model is trained with $E_t$ and evaluated with $E_v$. Data and training details are described in [A.5].

## 4.2 TYPE I: SINGLE CONTEXT, OVERLAPPING SAMPLE

Data elicited from one question during a collective response (CR) represents a type I primitive. We test type I model performance as a function of *vote per participant* (VPP) for two modes of CR; asynchronous and synchronous. In an *asynchronous* CR, participation happens on a rolling basis, so participants only vote on perspectives submitted prior to their participation. In a *synchronous* CR, all

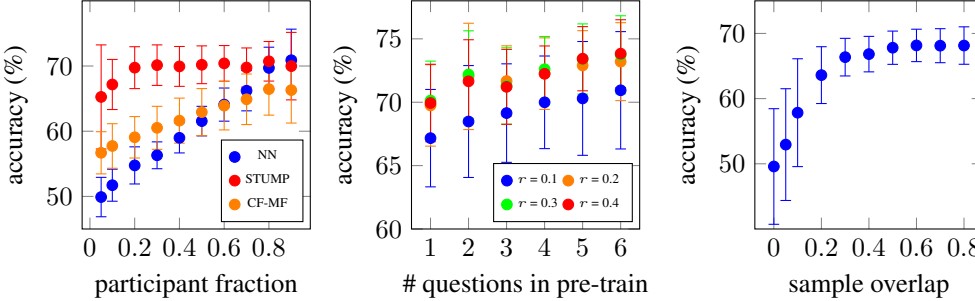

Figure 6: Type III inference accuracy vs participant fraction $r$.

Figure 7: Type III inference accuracy vs questions in pre-training data.

Figure 8: Type III/IV inference accuracy vs sample overlap.

participation happens in the same minute-scale window of time, so any perspective may be voted on by any participant.

**Type I as a baseline.** We simulate both cases while varying VPP by partitioning the appropriate votes into training and validation sets. To simulate the asynchronous CRS scenario, participants' votes pertaining to perspectives submitted after their own become the validation set $E_v$, and votes pertaining to perspectives submitted before their own become the training pool. We modulate vote density by adjusting the fraction of votes from the training pool which is randomly partitioned into the training set $E_t$. To simulate the synchronous CRS scenario, we randomly partition votes such that the training set $E_t$ and validation set $E_v$ are the same size as those in the asynchronous scenarios for each experiment. For each experiment, we train the models for each question and compute the corresponding validation accuracy.

STUMP outperforms all baselines at $< 30$ VPP and achieves a level of accuracy at only $\sim 10$ VPP which the NN baseline only achieves with 3x the amount of data (Figure 3). As the simplest elicitation primitive, Type I performance across simple baselines informs what constitutes *meaningful accuracy* in general. The *Random* baseline achieves $50\%$ accuracy (not shown) and the *Most Common* baseline saturates at an average of $\sim 64\%$ accuracy with a high degree of variance. A reasonable conclusion is that $50 - 64\%$ accuracy is not meaningful, $65 - 69\%$ is arguably meaningful, and $70\%$ or more is meaningful.

### 4.3 TYPE II: SINGLE CONTEXT, NON-OVERLAPPING SAMPLE

**Type I to Type II transition**. We simulate EI within a single-question context as the overlap between two samples decreases to zero. For each question, we randomly split the participants and perspectives into two equal sized samples: $\{H_{s1}, P_{s1}\}$ and $\{H_{s2}, P_{s2}\}$. Agreement votes by participants in $H_{s1}$ on perspectives in $P_{s2}$ become the validation set $E_v = \{e_{ij}^a, e_{ijk}^c \mid i \in H_{s1}, j, k \in P_{s2}\}$. The remaining votes contained in the question become the Type I training set $E_t$, where sample overlap is maximum, $E_t = \{e_{ij}^a, e_{ijk}^c \mid i \in H, j, k \in P_{s1}\} \cup \{e_{ij}^a, e_{ijk}^c \mid i \in H_{s2}, j, k \in P_{s2}\}$. Sample overlap is decreased by removing votes on perspectives in $P_{s1}$ by participants in $H_{s2}$ until none remain, giving Type II training set, $E_t = \{e_{ij}^a, e_{ijk}^c \mid i \in H_{s1}, j, k \in P_{s1}\} \cup \{e_{ij}^a, e_{ijk}^c \mid i \in H_{s2}, j, k \in P_{s2}\}$ (more experimental details are described [A.6.1]). We define sample overlap $p$ as the fraction of participants in $H_{s2}$ with training votes in $P_{s1}$, and compute validation accuracy for STUMP, NN and CF-MF baseline as $p$ varies (Figure 4). It is shown that NN and CF-MF baselines are sensitive to sample overlap. In contrast, STUMP's performance remains stable even at $p = 0$; *evidence of STUMP's ability to maintain latent-space alignment across non-overlapping samples.*

**Type II with Different languages**. We investigate STUMP's inference capability across two non-overlapping samples in different languages. Perspectives in one sample are translated into one of nine different languages using Google Translate API: German (de), Spanish (es), French (fr), Italian (it), Japanese (ja), Dutch (nl), Portuguese (pt), simplified Chinese (zh) and traditional Chinese (zhtw) [A.6.1]. We compute STUMP's Type II accuracy for each language pair and compare it to an English-English baseline (Figure 5). Performance across all language pairs is on par with the baseline.

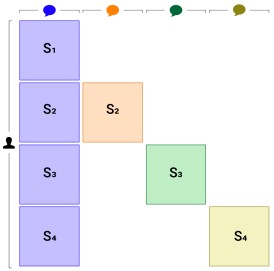

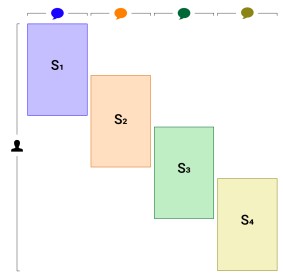

Figure 9: Case 1 sampling protocol, where shaded regions correspond to sampled votes and color corresponds to context.

Figure 10: Case 2 sampling protocol, where shaded regions correspond to sampled votes and color corresponds to context.

### 4.4 TYPE III: MULTIPLE CONTEXT, OVERLAPPING SAMPLE

**Transfer learning**. To test STUMP's capability for transfer learning, we pre-train participant embeddings in one context and use them for inference in a new context. Given a pair of contexts $Q_1$ and $Q_2$, we denote all perspectives in $Q_1$ as $P^1$ and all perspectives in $Q_2$ as $P^2$. We split the participants who participated in both contexts randomly into two groups $H_{s1}$ and $H_{s2}$. The pre-training set contains votes by all participants on perspectives in context $Q_1$, $E_{t1} = \{e_{ij}^a, e_{ijk}^c \mid i \in H, j, k \in P^1\}$, and is used to pre-train participant embeddings. The second training set contains votes for a fraction $r$ of participants on perspectives in a new context $Q_2$, $E_{t2} = \{e_{ij}^a, e_{ijk}^c \mid i \in H_{s2}, j, k \in P^2\}$, and $r = n(H_{s2})/(n(H_{s1}) + n(H_{s2}))$. The pre-trained embeddings are frozen while the transformation to latent space for the new context is learned. Votes on perspectives in $Q_2$ by participants not in the second training set comprise the type III validation set $E_v = \{e_{ij}^a, e_{ijk}^c \mid i \in H_{s1}, j, k \in P^2\}$ (more details are described in [A.6.2]). We vary the *participant fraction* $r$ from 0.05 to 0.9, and compute validation accuracy for STUMP and the CF-MF and NN baselines (Figure 6). STUMP performance increases with participant fraction when it is small, leveling off at $r \sim 0.2$, while the NN baseline only reaches similar performance around $r \sim 0.8$. The CF-MF baseline saturates at lower accuracy. In other words, STUMP can leverage data from a different context via transfer learning to achieve the same performance as the NN baseline with 75% fewer data.

**Increased pre-training data**. Next, we investigate how more pre-training data from more contexts affects STUMP's transfer learning performance. We increase the number of question contexts whose votes are included in the pre-training set [A.6.2] and compute type III validation accuracy at different participant fractions (Figure 7). STUMP's transfer learning performance increases as the number of question contexts included in pre-training increases.

### 4.5 TYPE IV: MULTIPLE CONTEXT, NON-OVERLAPPING SAMPLE

**Type III to Type IV transition**. Finally, we explore transfer learning across contexts as sample overlap decreases. Let *sample overlap* $p$ be the fraction of participants in the training sample (context $Q_2$) who are also in the pre-training sample (context $Q_1$). Moving from Type III to Type IV means decreasing $p$ from 1 to 0. To simulate this, we start with a Type III setup and iteratively remove participants from the pre-training sample who are also in the training sample (more experimental details are described in [A.6.3]). We use the training procedure and validation set from [4.4] and compute accuracy as $p$ varies (Figure 8). STUMP performance is arguably meaningful when sample overlap is larger than 50% but drops below that.

## 5 INCREASING DATA EFFICIENCY WITH SCALE

Based on STUMP's performance characteristics across elicitation primitives, we design two arbitrarily scalable elicitation protocols where: a) one may expect STUMP to sustain meaningful inference accuracy, and b) data leverage is positive, ie. $\frac{\partial \beta}{\partial n_t} > 0$.

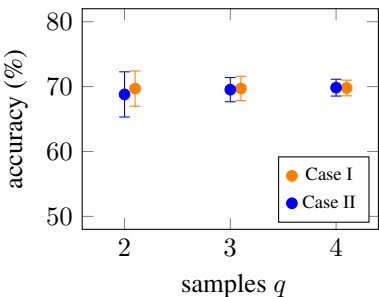
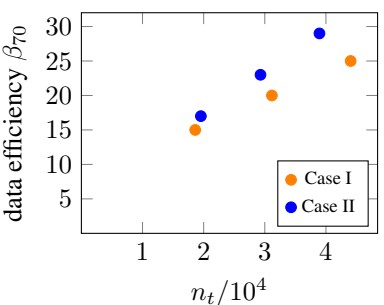

Figure 11: Inference accuracy as a function of the number of local samples.

Figure 12: Data leverage as a function of the number of votes in the training set.

**Case I**. STUMP shows meaningful type III inference for low participant fractions (Figure 6). This means if participants all vote on perspectives from a shared context, only a small fraction need to vote on perspectives from a new context to meaningfully infer the rest. Thus, the following elicitation protocol is viable: Elicit an initial sample where $m$ participants each give 1 new perspective and vote on $c$ other perspectives for a single context. Let each sample after the first be elicited from $m$ new participants where each a) provides 1 new perspective and votes on $c$ other perspectives for a new context, and b) votes on $c$ perspectives for the context shared by all samples (Figure 9). This protocol results in data efficiency of $\beta = n_t/(4c^2) + m/(2c) + m/(4n_t) \sim O(n_t)$, and data leverage $\frac{\partial \beta}{\partial n_t} \approx 1/(4c^2) > 0$. [A.7.1]

**Case II**. STUMP shows arguably meaningful type III/IV inference once sample overlap approaches $\sim 0.5$ (Figure 8). In other words, inference for a population of participants across two contexts requires around half of them to vote on perspectives from both contexts. This implies such an overlap supports latent space alignment between contexts. To maintain latent space alignment across an arbitrary number of contexts, we chain successive overlaps together with the following elicitation protocol: Elicit an initial sample where $m$ participants each provide 1 new perspective and vote on $c$ other perspectives for a single context. Let each sample after the first be elicited from $m$ participants where $m/2$ participants are new, and $m/2$ are from the previous sample. Let each participant provide 1 new perspective and vote on $c$ other perspectives for a new context (Figure 10). This protocol results in data efficiency of $\beta = n_t/(2c^2) + m/(2c) \sim O(n_t)$, and $\frac{\partial \beta}{\partial n_t} = 1/(2c^2) > 0$. [A.7.2]

**We simulate EI under both protocols** for $q = 2, 3, 4$ samples by partitioning the appropriate perspectives, participants, and votes into training and validation sets [A.8]. In both cases STUMP sustains a meaningful accuracy of $\sim 70\%$ as $q$ increases (Figure 11), approximating $O(n_t^2)$ votes using only $O(n_t)$. Thus, the combination of either elicitation protocol with STUMP achieves $\beta_{70} \sim O(n_t)$ (Figure 12), giving positive data leverage: $\frac{\partial \beta_{70}}{\partial n_t} = const > 0$. In other words, *data efficiency increases linearly with scale*.

## 6 DISCUSSION

**Improving inference accuracy.** This work prioritized data efficiency with scale over maximizing inference accuracy. We do not expect STUMP to achieve SOTA accuracy in its current form. Neural collaborative filtering (NCF), which replaces the inner product between latent space embeddings with a neural architecture able to learn a more general function, has been shown to improve performance in recommenders He et al. (2017), and is likely to improve accuracy in this setting as well (preliminary experiments have confirmed this). Experiments in (Section 4.4) indicate more, and more diverse, pre-training data can improve accuracy. STUMP's use of information in the perspective text is limited by the LLM it employs and we expect a SOTA LLM Chowdhery et al. (2022); Zhang et al. (2022); Mitchell et al. (2022) can improve accuracy. Other directions to improve accuracy may include: using participant side information like demographics, extending STUMP to use information in prompts, using active sampling to elicit votes expected to give the most information Gonzalez and Eltinge (2008); Zhang et al. (2020), and fine-tuning LLM weights while training STUMP. These will be our future work directions.

**Limitations.** STUMP fails to align latent spaces across non-overlapping samples from different contexts. This may require an embedding distribution alignment module and Stien Path training Liu et al. (2021). STUMP makes assumption of a low-rank structure and and does not model non-linearity. Our work does not quantify confidence in aggregate vote statistics, though Bilich et al. (2019) shows stochastic weight averaging Maddox et al. (2019) can estimate the posterior variance in this setting. We only use text perspectives. Handling perspectives spanning text, images, audio, and video requires swapping STUMP's LLM for a multi-modal foundation model able to generate aligned embeddings across media types Jabbar et al. (2022); Radford et al. (2021); Bommasani et al. (2021); Song et al. (2022); Reed et al. (2022). Our experiments were limited by data which was available, and it is possible that some kinds of contexts, perspectives, or participants are materially different in ways that make our results generalize less than expected. STUMP can only represent changing participant views by treating different times as different contexts, while recommenders often model it explicitly Chen et al. (2021a). This work does not show how to make results understandable for human agents, though clustering in latent space is a promising direction Small et al. (2021). We don't show how learned preference signals can directly control other ML models, but jury learning for classifiers Gordon et al. (2022) and reward learning for RL agents Ziegler et al. (2019) are encouraging directions. STUMP is not differentially private Dwork (2008) but approaches developed for LFMs in recommenders may be applicable Berlioz et al. (2015). Finally, elicitation inference is just one of the components helpful for achieving the underlying goals of multi-principal-agent alignment.

**Risks.** In CRS, some biases common to recommenders Chen et al. (2020) are less significant due to the structure of the task. In particular, if vote exercises are random and can't be skipped, while users are anonymous and can't see results, then biases spanning selection, exposure, conformity, position, and popularity are minimized. However, participants who respond and vote more slowly may still be under-represented in the data, bad elicitation prompts can still bias participants Litwak (1956), and malicious actors may still attempt to influence results through coordinated participation. Biases in pre-trained LLMs Schramowski et al. (2022); Nadeem et al. (2021); Liu et al. (2022) may also propagate through the model to effect inference in unknown ways. We avoid active sampling Zhang et al. (2020) due to the risk of compounding small model bias into large inference bias. Further, non-representative participant populations can arise from biased solicitation, or unequal willingness or ability to participate. As such systems become used for important decision-making, they risk neglecting views of already marginalized communities due to limited technology access and capacity Roessler (2018) unless specifically mitigated for, e.g. using existing inclusive voting infrastructure, structurally increasing digital equity, and/or sortition with high touch support. Even if the data collected is unbiased and representative, there are still risks that unexpected inference errors can corrupt results, or that even high-quality results may be misinterpreted or misused due to miscalibrated confidence, missing context, or malice. Elicitation inference more generally enables a different set of tradeoffs than traditional approaches to vote collection and aggregation. Its efficiency gains can enable increased access to agenda-setting power compared to a traditional vote—with the tradeoff that this involves accepting approximation and error bars in vote outcomes.

**Conclusion.** We have demonstrated EIO for collective response systems where data efficiency increases linearly with the size of the elicited data set. As a result, the number of elicitations needed per participant remains constant even as the number of perspectives and participants grows. Our experiments suggest this scaling relationship is likely to hold for data sets spanning contexts, languages, and population samples. If true, that would make it feasible to approximate high-complexity preference signals for massive principal populations spanning open-ended perspectives across an agent's decision domain. Such signals may be used to steer complex objective and evaluation functions for multi-principal-agent scenarios spanning governance, markets, diplomacy, and AI system alignment. For example, to support collective response for peacebuilding Alavi et al. (2022) and constructive democratic governance of global platforms Ovadya (in combination with other processes and tools). More speculatively, EIO may help ensure that the quality and complexity of human input driving the actions of powerful automated systems increases with the capabilities of those systems [A.9].

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

## A    APPENDIX

### A.1    ELICITATION FEASIBILITY

Assume a scenario where there are 10,000 principals and an agent with an open-ended decision space such that each principal may provide a unique perspective related to decisions the agent could make in a given month. Consider 10,000 perspectives, and assume that directly eliciting a principal's view on a perspective involves casting a single vote which takes 10 seconds each. Direct elicitation of all votes equates to 28 hours spent voting per principal per month ( 280,000 hours per month in total). This is unreasonable but theoretically possible. However, increasing this to just 100,000 principals equates to 280 hours per principal per month – more than a full-time job –, and increasing it to 1,000,000 principals equates to 2,800 hours per principal per month which is more than the number of hours in a month. There are more than 300,000,000 principals (US citizens) in the US government agent.

### A.2    SCALING BEHAVIOR OF EIO FOR CRS BASED ON PREVIOUS APPROACHES

Consider a human-perspective matrix $\Theta$ corresponding to $N$ humans and $K$ perspectives spanning $Q$ different contexts (ie. elicited by $Q$ different questions or prompts). Assume the corresponding CRS generates a finite number of $k$ perspectives for each context so $K = Qk$. The number of elements in $\Theta$ is $NQk \sim O(NQ)$. Bilich et. al. demonstrate an EIO approach to CRS in (Bilich et al., 2019) where only a finite number of votes per person, $c$, is needed to recover an approximation of $\Theta$ for a single context. This means the number of votes which need to be sampled for a $\Theta$ spanning $Q$ contexts is $NQc \sim O(NQ)$.

### A.3    NUCLEAR NORM MODEL DETAILS

The nuclear norm model likelihood is given by:

$$p(A, D, C \,|\, u) = \prod_{i,j \in A} \sigma(u_{ij}) \prod_{i,j \in D} (1 - \sigma(u_{ij})) \prod_{i,j,k \in C} \sigma(m_{ij} - m_{ik})$$

where $A = \{i, j | a_{ij}\}$, $D = \{i, j | d_{ij}\}$, and $C = \{i, j, k | c_{ijk}\}$. The NN model posterior is given by

$$p(u \,|\, A, D, C) = \frac{1}{Z} p(A, D, C \,|\, u) \cdot \mathbf{1}_{||u||_* < \tau}$$

$$Z := \int_{\mathbb{R}^{n \times m}} p(A, D, B \,|\, M) \cdot \mathbf{1}_{||M||_* < \tau} \mathrm{d}M.$$

where $\mathbf{1}_{||u||_* < \tau}$ denotes a uniform prior over the nuclear norm ball for $u$ of radius $\tau$. Leaning happens through optimization via SGD of the MAP estimator.

## A.4 DATA SOURCE

Data used for this research comes from periodic load testing done by a private company for internal quality assurance, and was fully de-identified prior to use in the research. The researchers identified that portions of it could be interesting for this work due to the scale of the data and the predictive challenges associated with the nuance and ambiguity of certain questions that were asked. Data made available for this research came from the following open-ended questions asked during the collective response load test:

- *During the pandemic, what fun experiences have you newly discovered? These can be any activities, hobbies, or events you consider fun.*
- *Please tell me more: what is more important to you now?*
- *And conversely, what is less important to you now?*
- *What new routines or behaviors have you started, if any, based on what you now think is important?*
- *How or where did you learn the new routine or behavior?*
- *Tell me about a product or service you have used MORE than ever in the past year. Why did you use it more?*
- *Please tell me more – how is this impacting your decisions, if at all?*

## A.5 DATA AND TRAINING DETAILS

For each question in the dataset, there are an average of 529 perspectives, 14455 agreement exercises, and 10151 pair-choice exercises. The full dataset consists of 172,242 votes across 3,703 perspectives made by 534 humans.

In training the STUMP model, we use $L2$ regularization on the transformation weight $\mathbf{M}$ and apply dropout on the universal embedding. We use a pre-trained multilingual universal sentence encoder (Apache-2.0 license) to generate perspective embeddings $F(p_i | \phi)$. Hyper-parameters are tuned by grid search, and the following values are used in our experiments: regularization rate = 0.01, dropout rate = 0.3, latent participant embedding dimensions = 50, and learning rate = 0.001 with the Adam optimizer. The models are implemented with PyTorch, and trained on a PC with 4-core Intel i7-1165G7 processor.

## A.6 ELICITATION PRIMITIVE EXPERIMENT DETAILS

The exercises in the training sets consist of a mix of agreement and pair-choice exercises where the agreement exercises account for 60%–75% of the total exercises.

### A.6.1 TYPE II: SINGLE CONTEXT, NON-OVERLAPPING SAMPLE

**Type I to Type II transition.** For each question, we randomly split the participants into two equal-sized groups $H_{s1}$ and $H_{s2}$, and their corresponding perspectives $P_{s1}$ and $P_{s2}$. We select votes on perspectives in $P_{s2}$ by participants in $H_{s1}$ as the validation set, $E_v = \{e_{ij}^a, e_{ijk}^c \mid i \in H_{s1}, j, k \in P_{s2}\}$. The train set consists of the following three pieces: votes on perspectives in $P_{s1}$ by participants in $H_{s1}$, votes on perspectives in $P_{s2}$ by participants in $H_{s2}$, and votes perspectives in $P_{s1}$ submitted by a fraction $p$ of participants in $H_{s2}$, where $p \in [0, 1]$, $E_t = \{e_{ij}^a, e_{ijk}^c \mid i \in$

$H_{s1}, j, k \in P_{s1}\} \cup \{e^a_{ij}, e^c_{ijk} \mid i \in H_{s2}, j, k \in P_{s2}\} \cup \{e^a_{ij}, e^c_{ijk} \mid i \in H\prime_{s2}, j, k \in P_{s1}, H\prime_{s2} \subset H_{s2}, |H\prime_{s2}|/|H_{s2}| = p\}$. When $p = 1$, the train data has maximum participant overlap, i.e., it contains max number of participants with votes on perspectives in both $P_{s1}$ and $P_{s2}$. When $p = 0$, the train data has zero participant overlap, i.e., it contains with votes on perspectives in both $P_{s1}$ and $P_{s2}$ (Figure 13a).

**Type II with Different languages.** For each question, we randomly split the participants into two groups $H_{s1}$ and $H_{s2}$, and their corresponding perspectives $P_{s1}$ and $P_{s2}$. The train set consists of exercises pertaining to perspectives in $P_{s1}$ submitted by participants in $H_{s1}$, and exercises pertaining to perspectives in $P_{s2}$ submitted by participants in $H_{s2}$, $E_t = \{e^a_{ij}, e^c_{ijk} \mid i \in H_{s1}, j, k \in P_{s1}\} \cup \{e^a_{ij}, e^c_{ijk} \mid i \in H_{s2}, j, k \in P_{s2}\}$. The validation set consists of exercises pertaining to perspectives in $P_{s1}$ submitted by participants in $H_{s2}$, and exercises pertaining to perspectives in $P_{s2}$ submitted by participants in $H_{s1}$, $E_v = \{e^a_{ij}, e^c_{ijk} \mid i \in H_{s2}, j, k \in P_{s1}\} \cup \{e^a_{ij}, e^c_{ijk} \mid i \in H_{s1}, j, k \in P_{s2}\}$. The second group perspectives are translated to nine languages respectively: German (de), Spanish (es), French (fr), Italian (it), Japanese (ja), Dutch (nl), Portuguese (pt), simplified Chinese (zh) and traditional Chinese (zhtw).

A.6.2   TYPE III: MULTIPLE CONTEXT, OVERLAPPING SAMPLE

**Transfer learning.** Given a pair of contexts $Q_1$ and $Q_2$, we denote all perspectives in $Q_1$ as $P^1$ and all perspectives in $Q_2$ as $P^2$. We split the participants who participated in both contexts randomly into two groups $H_{s1}$ and $H_{s2}$. The pre-train set contains exercises pertaining to perspectives in $P^1$ submitted by all participants, $E_{t1} = \{e^a_{ij}, e^c_{ijk} \mid i \in H, j, k \in P^1\}$. The second train set contains exercises pertaining to perspectives in $P^2$ submitted by participants in $H_{s2}$, $E_{t2} = \{e^a_{ij}, e^c_{ijk} \mid i \in H_{s2}, j, k \in P^2\}$. The validation set contains exercises pertaining to perspectives in $P^2$ submitted by participants in $H_{s1}$, $E_v = \{e^a_{ij}, e^c_{ijk} \mid i \in H_{s1}, j, k \in P^2\}$. Participant fraction $r$ is defined as $r = n(H_{s2})/[n(H_{s1}) + n(H_{s2})]$ (Figure 13b).

The model is pre-trained using the pre-train set $E_{t1}$, and the learned participant embeddings $\mathbf{z}$ are transferred to the next training cycle using the second train set $E_{t2}$. In the second training cycle, $\mathbf{z}$ is fixed while the new context transformation $\mathbf{M}^2$ is learned. Then the participant embeddings $\mathbf{z}$ learned from pre-training and context transformation $\mathbf{M}^2$ learned from the second training are together used to make predictions for exercises in the validation set.

**Increased pre-training data.** Given a target context $Q_t$ and its perspectives $P^t$, we select a set of $N - 1$ contexts that do not contain $Q_t$ as the pre-train contexts. We split the participants who participated in all of the selected questions randomly into two equal-sized groups $H_{s1}$ and $H_{s2}$. The pre-train set contains exercises pertaining to perspectives in the pre-train contexts submitted by all participants, $E_{t1} = \cup_{n=1...N, n \neq t}\{e^a_{ij}, e^c_{ijk} \mid i \in H, j, k \in P^n\}$. The second train set contains exercises pertaining to perspectives in $P^t$ submitted by participants in $H_{s2}$, $E_{t2} = \{e^a_{ij}, e^c_{ijk} \mid i \in H_{s2}, j, k \in P^t\}$. The validation set contains exercises pertaining to perspectives in $P^t$ submitted by participants in $H_{s1}$, $E_v = \{e^a_{ij}, e^c_{ijk} \mid i \in H_{s1}, j, k \in P^t\}$.

A.6.3   TYPE IV: MULTIPLE CONTEXT, NON-OVERLAPPING SAMPLE

**Type III to Type IV transition.** Given a pair of contexts $Q_1$ and $Q_2$ and their corresponding perspectives $P^1$ and $P^2$, we split the participants who participated in both contexts randomly into two equal-sized groups $H_{s1}$ and $H_{s2}$. The pre-train set contains exercises pertaining to perspectives in $P^1$ submitted by participants in $H_{s1}$ and exercises pertaining to perspectives in $P^1$ submitted by a fraction $p$ of participants in $H_{s2}$, where $p \in [0, 1]$, $E_{t1} = \{e^a_{ij}, e^c_{ijk} \mid i \in H_{s1}, j, k \in P^1\} \cup \{e^a_{ij}, e^c_{ijk} \mid i \in H\prime_{s2}, j, k \in P^1, H\prime_{s2} \subset H_{s2}, |H\prime_{s2}|/|H_{s2}| = p\}$. The second train set contains exercises pertaining to perspectives in $P^2$ submitted by participants in $H_{s2}$, $E_{t2} = \{e^a_{ij}, e^c_{ijk} \mid i \in H_{s2}, j, k \in P^2\}$. The validation set contains exercises pertaining to perspectives in $P^2$ submitted by participants in $H_{s1}$, $E_v = \{e^a_{ij}, e^c_{ijk} \mid i \in H_{s1}, j, k \in P^2\}$. When $p = 1$, $E_{t1}$ and $E_{t2}$ have maximum participant overlap. When $p = 0$, they have zero participant overlap (Figure 13c).

Figure 13: Elicitation primitive experimental setup

(a) Type I to Type II transition      (b) Type III      (c) Type III to Type IV transition

## A.7 DATA EFFICIENCY SCALING FOR ELICITATION PROTOCOLS

### A.7.1 CASE I

The first sample elicits $mc$ votes and initializes $\Theta$ with $N = m$ participants and $K = m$ perspectives. Each additional sample elicits an additional $2mc$ votes and adds $m$ new participants and $m$ new perspectives to $\Theta$. After $q$ samples, $n_t = mc(2q-1)$ votes have been elicited, and $\Theta$ is comprised of $K = mq$ perspectives and $N = mq$ humans resulting in $n(\Theta) = m^2q^2$. Noting the relation $q = (n_t + mc)/(2mc)$, then $n(\Theta) = n_t^2/(4c^2) + (n_t m)/(2c) + m/4 \sim O(n_t^2)$, data leverage is $\beta = n_t/(4c^2) + m/(2c) + m/(4n_t) \sim O(n_t)$, and $\frac{\partial \beta}{\partial n_t} \approx 1/(4c^2)$ for $n_t >> \sqrt{m}/2$.

### A.7.2 CASE II

The first sample elicits $mc$ votes and initializes $\Theta$ with $N = m$ participants and $K = m$ perspectives. Each additional sample elicits an additional $mc$ votes and adds $m/2$ new participants and $m$ new perspectives to $\Theta$. After $q$ samples, $n_t = mcq$ votes have been elicited, and $\Theta$ is comprised of $K = mq$ perspectives and $N = (m/2)(q+1)$ humans resulting in $n(\Theta) = (m^2q^2 + m^2q)/2$. Noting the relation $q = n_t/(mc)$, then $n(\Theta) = n_t^2/(2c^2) + (n_t m)/(2c) \sim O(n_t^2)$, data leverage is $\beta = n_t/(2c^2) + m/(2c) \sim O(n_t)$, and $\frac{\partial \beta}{\partial n_t} = 1/(2c^2)$

## A.8 PROTOCOL SIMULATION EXPERIMENTS

**Case I** Given a set of perspectives corresponding to $Q$ different contexts $P^1...P^Q$, and $Q$ equal sized non-overlapping sets of participants $H_{s1}...H_{sQ}$, we construct the following experiment for each $q = 2...Q$: let $\Theta^q$ be the HPM corresponding to $\cup_{n=1}^q \{H_{sn}\}$ and $\cup_{n=1}^q \{P^n\}$. Partition the training set in each case to correspond to local samples denoted by the shaded areas in Figure 9 such that $E_t^q = \cup_{n=2}^q \{e_{ij}^a, e_{ijk}^c \mid i \in H_{sn}, j, k \in P^n\} \cup \{e_{ij}^a, e_{ijk}^c \mid i \in \cup_{n=1}^q H_{sn}, j, k \in P^1\}$. Let the validation set $E_v$ contain all other exercises in $\Theta^q$, denoted by the white areas in Figure 9.

**Case II** Given a set of perspectives corresponding to $Q$ contexts $P^1...P^Q$, and $Q$ equal sized partially overlapping sets of participants $H_{s1}...H_{sQ}$ with adjacent overlap $p$ defined by $|H_{si} \cap H_{si+1}| = p * |H_{si}|$, we construct the following experiment for each $q = 2...Q$: let $\Theta^q$ be the HPM corresponding to $\cup_{n=1}^q \{H_{sn}\}$ and $\cup_{n=1}^q \{P^n\}$, the train set is denoted by the shaded areas in Figure 10, $E_t^q = \cup_{n=1}^q \{e_{ij}^a, e_{ijk}^c \mid i \in H_{sn}, j, k \in P^n\}$, and the validation set $E_v$ contains all other exercises in $\Theta^q$, denoted by the white areas in Figure 10.

## A.9 SPECULATION ON FOUNDATION MODEL ALIGNMENT SCENARIO

As foundation model (FM) capabilities increasingly span LLMs and RL agents Bommasani et al. (2021); Reed et al. (2022); Chen et al. (2021b); Zheng et al. (2022); Janner et al. (2021), sharing an FM between an EIO model and an RL agent may enable a scenario where the quality and nuance of

normative alignment signals improves as the agent learns. As an example, this could happen in a scenario where the following are true:

1. The quality of perspective representations (e.g. embeddings) produced by the shared FM, and used by the EIO model, improves as the RL agent learns.
2. The performance (e.g. inference accuracy or data efficiency) of EIO increases as the quality of perspective representations improves.

## A.10 PARTICIPANT PAYMENT AND CONSENT

The 534 participants who joined the load testing exercise had been solicited through online panel providers for an incentive of approximately $15/hour USD, and consented to the policy shown in Figure 14. While this consent form does mention PII, this was a general consent form for the data provider and no PII was known to be contained in the data set made available for this research. The researchers were structurally limited from accessing any directly collected PII due to system design, and none of the raw outputs that were inspected directly during the research included any PII.

Figure 14: Screenshot of consent policy presented to all participants as part of the load test.

