# OpenReview forum: "Elicitation Inference Optimization for Multi-Principal-Agent Alignment"
_ICLR.cc/2023/Conference — Submitted to ICLR 2023_

### Official Review · Reviewer_wpVc · 2022-10-25

**Confidence:** 2
**Correctness:** 3
**Technical Novelty And Significance:** 4
**Empirical Novelty And Significance:** 4
**Recommendation:** 5

**Clarity, Quality, Novelty And Reproducibility:**

Clarity/Reproducibility

+ As I mention above, the paper needs to explain more details. Currently the empirical section has very sparse details, and just describes results. This makes it very hard for the reader.

Novelty

+ From my best guess judgement, the approach proposed to this problem seems new. The problem is a challenging one and the authors explain each of their components in an intuitive fashion. Also, it seems to have very little direct related work which by definition makes this paper novel.

**Strength And Weaknesses:**

Strength

+ I personally found the problem setting to be very interesting. The paper was a bit hard to parse, but an example provided in the paper helps. The goal of the problem is to predict whether a user will agree with a perspective in dialogue systems. An example is when a user is asked "What would you say if you saw your parents smile?" would they agree with the perspective ""I feel happy". These questions are very important for designing interactive real-world systems and thus, designing good algorithms that are robust in a wide variety of datasets is important.

+ The paper overall does extensive empirical validation and performs quite competitively against the one non-trivial baseline from prior work in a number of setting. To that end, it seems to show good results. However, I am a bit worried that the other two chosen baselines are pretty weak and not a good comparision except that it provides a good sanity check.


Weakness

+ My primary qualm with the paper is that its pretty poorly written. The paper introduces a number of concepts in section 2 and then barely uses them in the paper. I think its better to introduce the terms at the location where they are used, if they are going to be used only once or twice. Another worry is that the empirical setup is not well explained. Its not clear to me what the model was trained on, what was the dataset, what were the inputs, how was the training and evaluation done, etc. The paper directly jumps to talking about the results. The empirical section is very hard to read. I believe more details needs to be added in that section so that the reader can verify the setup and details. From a reproducibility stand-point, its unclear how to set this experiments up if someone chose to.

+ Overall, the paper also needs to add more motivations for the problem. Although the problem is interesting technically, adding a concrete application would greatly benefit the reader. Since the authors compare this problem to collaborative filtering, I am surprised that the baselines chosen in the experiments do not have anything from the recommendations literature. Since there is a 1:1 correspondence between the two problems, is there a possibility of comparing against stronger baselines from the rec-sys literature?

**Summary Of The Paper:**

This paper considers the problem of Elicitation inference optimization. At a high-level, in this problem the goal is to get the perspectives of N participants on K different perspectives by minimizing the total number of elicitations. Elicitation could be direct: does participant i agree with perspective j or comparative, does participant i agree with perspective i over j. In this work they study this in the context of collective dialogue systems where the set of perspectives is an "open-world" meaning that super-set of possible perspectives is unknown apriori. The perspectives here are text-based; the paper proposes an algorithm that leverages logistic utility models with large-language models. They run sample experiments to show that this model can outperform random baseline, most-common baseline and nuclear norm baseline from prior work.

**Summary Of The Review:**

Overall, this paper has enough interesting aspects to get my interest. I have read this as carefully as possible, but still overall have a few things unanswered. In particular, the ones I have mentioned in the weakness. I believe this paper needs a major rewriting which would be needed for me to also understand the paper even better to judge the paper's correctness. At this point, I made some best guess judgement on how the experiment is setup. I also think, the authors should add some stronger baselines from collaborative filtering line of work and make the comparison to CF grounded in data.

---

### Official Review · Reviewer_Pf2b · 2022-10-26

**Confidence:** 2
**Correctness:** 4
**Technical Novelty And Significance:** 4
**Empirical Novelty And Significance:** 4
**Recommendation:** 6

**Clarity, Quality, Novelty And Reproducibility:**

The paper is mostly clear. I am not super familiar with the related literature but the results seem novel to me.

**Strength And Weaknesses:**

Strengths:
The paper proposes an unconventional method that improves the data efficiency of the existing EIO methods. The idea of integrating pre-trained LLM is novel and meaningful. Their model allows context-specific learning with a shared LLM across different contexts, which transfers learning from one context to other contexts. The model is evaluated comprehensively in four primitives, which gives insights into the performance of their approach. They also provide two novel elicitation protocols based on their empirical evaluation results,  and they show by both theoretical analysis and empirical simulations that their data efficiency increases linearly with scale.

Weaknesses:
I'm not super familiar with the literature and I find no glaring weakness in the paper.

**Summary Of The Paper:**

The paper studies the problem of approximating N principal’s views across K perspectives (which can be represented as a N*K matrix) with a minimum number of direct elicitation. They improve the data efficiency (N*K/# of direct elicitation) of the existing approaches by transferring learning from one context to support inference in other contexts and leveraging the information in the perspective text. They propose a method that becomes increasingly data-efficient as the amount of data elicited grows. They first introduce a model that integrates a pre-trained LLM with a latent factor model. They empirically evaluate the performance of their model across a range of elicitation primitives arising from two data set properties: context homogeneity and sample overlap. Finally, based on the evaluation, they design two elicitation protocols that guarantee meaningful accuracy while data efficiency increases linearly with scale.

**Summary Of The Review:**

Overall, I think this is a good paper with meaningful conceptual and methodological contributions, as well as insightful empirical evaluations. They innovatively improve the existing method by adopting LLM technologies. They also provide extensive empirical evaluations that give insights into the performance of their methods.

---

### Official Review · Reviewer_p4dW · 2022-10-29

**Confidence:** 4
**Correctness:** 3
**Technical Novelty And Significance:** 2
**Empirical Novelty And Significance:** 3
**Recommendation:** 5

**Clarity, Quality, Novelty And Reproducibility:**

I found the paper hard to follow at times, particularly the experimental details. In particular, more details should have been provided about the type of questions asked, how people vote on them, and how the experimental setup was constructed. Regarding the last point it was not clear to me whether the sample selection was done uniformly at random. If so, how was it ensured that all the overlap properties are maintained?

In terms of novelty, I think the idea of using pre-trained LLM for generating latent embedding is interesting. However, the setup looks very similar to low-rank matrix factorization problem with non-linear likelihood functions. Additionally, the choice of the utility functions is also quite simple IMO.

**Strength And Weaknesses:**

Strength: I think the idea of using pre-trained LLM for generating task embeddings is quite interesting. Additionally, the results on transfer learning looks promising as it can leverage samples across different contexts.

Weaknesses:
- The authors implicitly makes assumption of a low-rank structure. In particular the formulation of utility $u_{ij} = z_i \cdot w_j$ implies that the proposed method needs to only learn $O(N+K)$ parameters. In that respect the proposed method is not very different than matrix factorization based methods.
- The second drawback about the model is that the utility model assumes a specific structure ($P(a_{ij} | u_{ij}) = \sigma(u_{ij})$) which is quite restricted in my opinion. There are more general models of preferences e.g. Thurston-Mosteller or Random utility model. The authors also did not motivate the choice of the particular utility model.
- Perhaps the biggest weakness of the paper is the benchmark against which it is compared. Since the proposed method essentially learns low-dimensional embeddings for each user and perspective, the approach is similar to low-rank matrix factorization, and methods such as deep matrix factorization should have been compared. Additionally, both the proposed method and alternative nuclear norm based method are optimizing similar log-likelihood functions. This makes me think that STUMP does better only because it learns embeddings from pre-trained LLM.


**Summary Of The Paper:**

This paper considers the problem of eliciting $N$ principals views across $K$ topics. In particular, the goal is to ask significantly less than $NK$ questions and infer the remaining entries from such questions. The authors use a factored model where they learn a representation $z_i$ for the $i$-th principal and $w_j$ for the $j$-th topic. The utility of principal $i$ on perspective $j$ is $u_{ij} = z_i \cdot w_j$ and the response are assumed to follow sigmoid of these utility functions. Moreover, in order to learn the representation of each perspective, the authors use a pre-trained LLM to generate text embedding $e_j$ and then $w_j$ is set to tanh activated affine transformation with parameter $M$. The matrix $M$ and the parameter $\{z_i\}$ are learned by maximizing a likelihood function on the observed data.

The experiments compare the proposed method (STUMP) with a nuclear norm based method on datasets with two types of contexts and different overlap assumptions. The authors found that the proposed method performs well across all language pairs for the single context setting. For the multiple context setting, the authors showed that STUMP can reduce required amounts of data by leveraging data from a different context.

**Summary Of The Review:**

The paper introduces an interesting idea of using pre-trained LLM to generate embeddings for different perspectives which are then used to generate opinions of experts on missing entries. However, the choice of the utility function is simple, and the method is also not compared against a set of different alternative algorithms.

---

### Official Review · Reviewer_pwkh · 2022-12-07

**Confidence:** 3
**Clarity, Quality, Novelty And Reproducibility:** Pls see in the Weakness above.
**Correctness:** 3
**Technical Novelty And Significance:** 1
**Empirical Novelty And Significance:** 2
**Recommendation:** 3

**Strength And Weaknesses:**

Strength -

- Interesting and practically relevant problem setup. The problem of matrix completion or collaborative filtering from preference or reward feedback has many applications.

-  The idea of using pre-trained LLM for generating task embeddings might be interesting.

- Extensive evaluations

Weakness -

- The paper is extremely hard to read, and uses too many acronyms (which I found at times unnecessary, say EIO, CRM etc) and hinders the flow of reading/ affects the comprehensibility of the paper.

- Clarity: Notations are not well defined always:, e.g. E_t, E_v which are essentially the training and validation component of the main dataset E. I even found it hard to understand the problem setting and objective (beta_acc in Pg2) - it essentially is a fraction of total possible query to #required queries, so higher the accuracy better the performance, but it is written in a convoluted way in my opinion. Even the algorithm, embedding technique is not very well defined, I struggled to figure out the section/ paragraph which is dedicated for these details.

- Novelty of the paper is not clear. On a high level, the work used a pre-trained LLM to generate text embedding and used Frobenius norm-based likelihood maximization to fit the model parameters -- such techniques ar fairly well studied. I fail to see if there is any new key idea offered in the current methods.

- The paper did not discuss the theoretical performance of the suggested method, or compare that with any of the existing baselines.

- Experiments: the paper studies different experimental setups, but they do not seem to be reproducible since the settings are poorly defined, the used baselines (random, most-common, nuclear-norm-based optimization) also seem to be restricted as the literature of collaborative filtering is diverse and many recent works used more sophisticated techniques, which should have been compared against.

**Summary Of The Paper:**

This paper considers the problem of Elicitation inference optimization where the goal is to get the perspectives of N participants on K different perspectives by minimizing the total number of elicitations. Elicitation could be direct: does participant i agree with perspective j or comparative, does participant i agree with perspective i over j.

Authors study the problem in the context of collective dialogue systems where the set of perspectives is an open-world meaning that super-set of possible perspectives is unknown apriori. The perspectives here are text-based; the paper proposes an algorithm that leverages logistic utility models with large-language models. In particular, they use a factored model where they learn a representation z_i for the i-th principal and w_j for the j-th topic. The utility of principal on perspective is u_ij = z_i w_j  and the response are assumed to follow sigmoid of these utility functions. In order to learn the representation of each perspective, the authors use a pre-trained LLM to generate text embedding e_i and then w_j is set to tanh activated affine transformation with parameter M. The matrix M and the parameter z_i are learned by maximizing a likelihood function on the observed data.

They empirically evaluate the performance of their proposed method (STUMP) against existing baselines, across a range of elicitation primitives arising from two data set properties: context homogeneity and sample overlap. Baselines used were random baseline, most-common baseline and nuclear norm baseline, on datasets with two types of contexts and different overlap assumptions. The proposed method is claimed to become increasingly data-efficient as the amount of data elicited grows. They first introduce a model that integrates a pre-trained LLM with a latent factor model. Finally, based on the evaluation, they design two elicitation protocols that guarantee meaningful accuracy while data efficiency increases linearly with scale.

**Summary Of The Review:**

The problem setting could be interesting but already well-studied in the collaborative filtering/ matrix completion + representation learning  based literature. It is hard to appreciate the technical novelty of the paper. Moreover, the paper is very hard to parse and uses too many acronyms, and incomprehensible at time (some expt setup, theoretical novelty again existing baselines, etc). Repositioning the paper highlighting its key novelties (new contributions of this work), emphasizing on the reproducibility of the experiments, would greatly benefit the paper.

---

### Decision · Program_Chairs · 2023-01-20

**Decision:**

Reject

**Justification For Why Not Higher Score:**

The paper is too preliminary to be accepted: very hard to read, unclear novelty, and no theory.

**Justification For Why Not Lower Score:**

N/A

**Metareview: Summary, Strengths And Weaknesses:**

This paper studies elicitation inference optimization where the goal is to get K different perspectives of N participants while minimizing the total number of elicitations. There is no theory but the approach is empirically evaluated.

The paper got three borderline reviews and one additional review after the rebuttal. Every paper was supposed to get four reviews initially. However, one reviewer never submitted the review, even after my multiple attempts to reach out to them. I checked the three reviews before the rebuttal and they seemed sufficiently informative. However, during the discussion after the rebuttal, I noted that the reviewers are not as active as on other papers. This is when I realized that no reviewer is an expert on preference elicitation and I invited a trusted reviewer who published on this topic.

Based on the initial reviews, I marked this paper as a borderline and organized an online meeting with all reviewers to discuss it, including the new reviewer. This meeting happened on December 6 and the new reviewer submitted their review on December 7. I also looked at your paper before the meeting. This is a summary of the discussed concerns:

* Very hard to read. Even the algorithm is not clearly stated. This was one reason for low confidences of the reviewers.

* Unclear novelty. The optimized objective is a simple combination of one pairwise feedback loss and two absolute feedback looses. It is reminiscent of low-rank matrix factorization for collaborative filtering.

* No theoretical results. This is despite the fact that there are many in active learning.

* Hard to reproduce experiments with missing collaborative filtering baselines.

The recommendation to SAC was a result of half an hour discussion and not driven by a single review.

**Summary Of Ac-Reviewer Meeting:**

All points in the meta-review were discussed and the rejection vote was unanimous.